# Does a Rural-Urban Gradient Affect Beetle Assemblages in an Arid Ecosystem?

Mahmoud S. Abdel-Dayem [1,2,*], Mostafa R. Sharaf [1], Jonathan D. Majer [3], Mohammed K. Al-Sadoon [4], Ahmed M. Soliman [1], Abdulrahman S. Aldawood [1], Hathal M. Aldhafer [1] and Gamal M. Orabi [5]

1 Plant Protection Department, College of Food and Agricultural Sciences, King Saud University, Riyadh 11451, Saudi Arabia
2 Entomology Department, Faculty of Science, Cairo University, Giza 12613, Egypt
3 School of Biological Sciences, University of Western Australia, Perth, WA 6009, Australia
4 Department of Zoology, College of Science, King Saud University, Riyadh 11451, Saudi Arabia
5 Zoology Department, Faculty of Science, Suez Canal University, Ismailia 41522, Egypt
* Correspondence: mseleem@ksu.edu.sa or msabdeldayem@sci.cu.edu.eg; Tel.: +966-538-145-287

**Abstract:** Urbanization affects all elements of the pre-urban environment, including soils, hydrology, vegetation, and microclimate. Recently, Saudi Arabia has experienced rapid urbanization and growth. Thus, the country's biodiversity has been threatened. In the Riyadh region, beetle assemblages were assessed along a rural-suburban-urban gradient. A total of 2791 individuals from 94 species belonging to seven families were collected at 15 sites along three different gradients of urbanization in Wadi Hanifa, which runs for a length of 120 km from northwest to southeast. Tenebrionidae dominated abundance (60.1%) and richness (38%). Beetle abundance, evenness, and diversity were not different among habitats; however, species richness was higher in rural habitats. Detrended correspondence "DCA" and canonical correspondence "CCA" analyses showed distinct differences among sites along gradients. Urbanization intensity, soil variables, and land cover were significantly correlated with CCA axis 1, while elevation and flora were significantly correlated with CCA axis 2. The most critical operating environmental variables in Wadi Hanifa were buildings, elevation, soil organic carbon, litter cover, and litter depth, as well as plant species such as *Launaea capitata*, *Lycium shawii*, *Alhagi graecorum*, and *Heliotropium currasavicum*. Ten species in our study were associated with urban habitats, six with suburban habitats, and seven with rural habitats. Consequently, expanding urban areas may negatively affect the richness and composition of beetles and may result in the loss of some native species.

**Keywords:** abundance; Coleoptera; diversity; indicator species; Riyadh; species richness; urbanization; Wadi Hanifa





## 1. Introduction

Urbanization is not a recent phenomenon; the earliest urban settlements were first developed in eastern Asia, Mesopotamia, and Egypt. Urbanization gradually spread to all continents over time [1,2]. City dwellers have increased in number [3] with more than half of the global population now living in urban areas [4]. In cities, dense human populations associated with transportation activities and intensive industrial activity [5,6] results in significant landscape changes both at the large scale and within individual sites [7,8]. The consequences of city growth are noticeable on a landscape level as built-up areas disturb surrounding natural habitats producing partially or tiny isolated patches [9,10]. Additionally, the remnant natural habitat is often transformed into non-permanent and disturbed habitat, mainly abandoned sites, such as landfills, brownfields, gravel and sand pits, railway lands and industrial dumps, all of which are characterized by altered soil chemical and physical properties [11–13].

The effects of pre-urban development on soils, hydrology, vegetation, climate and animal populations can be seen in the altered regional environmental factors such as increased levels of nutrients, soil impermeability and pollution [8,14,15]. These conditions are expected to differentially affect species within the original communities by either favoring well-adapted species or displacing less adapted ones [16–18]. Such a selection process may lead to population decline or replacement for some native taxa with more generalist and tolerant individuals taking advantage of an improved environment [16,19]. The displacement of natives is usually associated with urbanization [13].

Invertebrates, specifically arthropods, make ideal study subjects for investigating urban biodiversity [2,20,21]. Their small size and environmental needs allow them to thrive in an urban environment, and their diverse life histories provide insight into the composition of metropolitan fauna [21,22]. Many studies have revealed that complex arthropod communities can be found within cities [17,18]. Beetle species are a significant part of Earth's animal population, representing about one-fourth of all known species [23], and many are threatened by human activity such as development or pollution [24]. Knowing how these populations change with respect to both abundance and composition of beetles allows ecologists and municipalities to monitor changes in the cityscape [2,24].

Since the 1970s, Saudi Arabia's population has increased rapidly as a result of oil revenues (3.4% annually) [25]. Consequently, large areas of formerly wild lands have been dramatically developed, most notably in the Al Sarawat, Hijaz and Northern and Central regions [25]. Consequently, some indigenous species that used to inhabit these areas are now at high risk of complete loss due to urbanization. For that reason, more ecological work needs to be done in these settings [26–28]. The aim of this study was to assess the environmental impact of urbanization in Wadi Hanifa (WH) (Central Saudi Arabia) along an urbanization gradient (rural, suburban and urban sites) using beetles as bioindicators.

## 2. Materials and Methods

### 2.1. Study Area

In the Wadi Hanifa (WH) in the Riyadh region of central Saudi Arabia, beetle diversity was investigated along a gradient of urbanization (Figure 1). As an eminent natural landmark in central Saudi Arabia, the WH occupies a geographical area between 24°30′ N–46°30′ E to 24°45′ N–46°45′ E. The wadi is 120 km long, with a depth ranging from 10 to 100 m and a width ranging from 100 to 1000 m. It stretches through the heart of Riyadh City from the northwest to the open desert in the southeast. Seasonal rains and sewage are the primary water sources in WH over more than 4000 square kilometers of the catchment area.

In terms of climate, WH has a relatively mild winter and a hot summer, with an average annual temperature of 26 °C and an average relative humidity of 24.4%. Annual precipitation is 85 mm, and the dry season, with the absence of rain, occurs from June to September [29].

In the rural area, sampling was conducted near Haysiah Dam, west of Al Uyaynah District (Diriyah Governorate) and 60 km northwest of the center of Riyadh City. The area included native and natural flora, such as *Acacia* spp. trees, and shrubs like Arabian boxthorn *Lycium shawii* Roem. & Schult. (Solanaceae) and harmel *Rhazya stricta* Decne. (Apocynaceae), and annual herbs such as huwa *Launaea capitata* (Spreng.) Dandy (Asteraceae) and little mallow *Malva parviflora* L. (Malvaceae). A number of unpaved roads are located in the area, as well as sheds for goats, sheep, and camels.

Compared to rural and urban habitats, the suburban area was characterized by deciduous casuarina trees *Casuarina equiseifolia* L. (Casuarinaceae), drought-tolerant species like camel thorn shrubs *Alhagi graecorum* Boise. (Fabaceae) and desert thistle *Echinops spinosissimus* Turra (Asteraceae), perennial herbs such as thumam *Pennisetum divisum* (Forssk. ex J.F.Gmel.) Henrard (Gramineae) and annual herbs such as rough cocklebur *Xanthium strumarium* L. (Asteraceae). A number of detached buildings and gardens, as well as paved and unpaved roads, were found in the area.

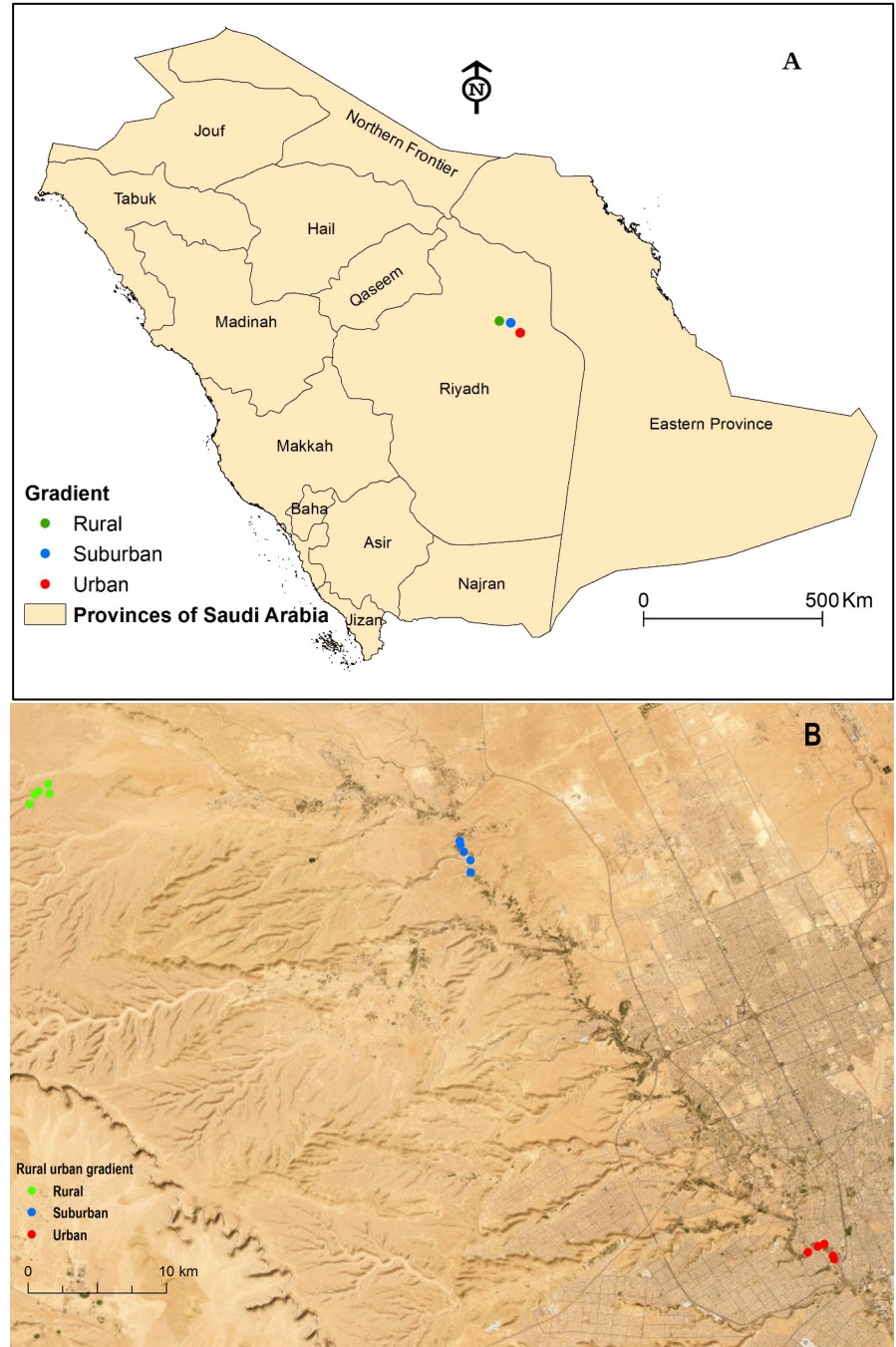

**Figure 1.** Map of the study area: (**A**) Within Saudi Arabia showing the study sites (colored dots) in Wadi Hanifa in Riyadh Province; (**B**) An enlarged representation of the sites in Wadi Hanifa along a continuum of rural-urban gradient (Rural, green dots; Suburban, blue dots; Urban, red dots).

Urban sampling sites were selected in the extensive park areas in Al Jaradiyah, Siyah, and Sultanah districts (Riyadh Governorate), 35 km southeast of suburban sampling sites. There are still patches of original vegetation on these sites, although they have been enriched by exotic plants and trees. The city's stormwater drainage system discharges groundwater into a surface flow channel that runs through the urban area until south of Al-Hair Town. Continuous water flows have created ideal conditions for diverse plant species to flourish (for instance, algae and other aquatic plants grow in abundance), wildlife communities to thrive, and recreational opportunities to flourish [14,30]. Several tree species can be found in the area, including *Acacia* spp., Prosopis, *Prosopis koelziana* Burkart,

and *P. juliflora* (Sw.) DC. (Fabaceae), Christ's thorn jujube, *Ziziphus spina-christi* (L.) Desf. (Rhamnaceae), and river red gum, *Eucalyptus camaldulensis* Dehnh. (Myrtaceae). There are also annual herbs such as lagoon saltbush, *Atriplex suberecta* I. Verd. (*Amaranthaceae*), and giant pigweed, *Trianthema portulacastrum* L. (Aizoaceae) as well as flowering perennial evergreen shrubs such as heliotrope, *Heliotropium currasavicum* L. (Boraginaceae), and Chaste shrub *Vitex* spp. (Lamiaceae).

### 2.2. Sampling Procedure

We conducted quantitative sampling following the protocols for the GLOBENET (Global Network for Monitoring Landscape Change) project. We randomly selected 15 sites (replicates) along a rural-suburban gradient (Figure 1). Each gradient contained five replicate sites comprising two parallel rows of five pitfall traps (plastic cup, diameter 10 cm, 15 cm depth and 5 m apart). A total of 150 pitfall traps were used in three gradients ($3 \times 50$). A distance of at least 200 m was maintained between sites to ensure sample independence. A 250-mm propylene glycol 40% mixture was used to fill the traps. Each trap was left working for a week, and sampling was conducted quarterly (January, April, July and October) for 12 months. The identification of the collected beetle species was conducted at the King Saud University Museum of Arthropods (KSMA), Department of Plant Protection, College of Food and Agricultural Sciences.

To measure the level of urbanization, we used the total amount of built-up area as a proxy. Buildings, pavements, roads, and asphalt-covered paths were the most influential determinants of built-up areas. A high-resolution aerial image captured in Google Earth was used to extract the determinants within each site for a square of one kilometer surrounding the sampling location (Table 1). We used a handheld GPS unit (Garmin, Montana 650 handheld Global Positioning System) to determine the elevation at each site.

**Table 1.** Means or ranges of variables measured at the three urbanization levels.

| Environmental Variable | Rural | Suburban | Urban |
|---|---|---|---|
| Elevation range (m) | 800–820 | 690–710 | 570–590 |
| Average Buildings (area in km$^2$) | 0 | 0.0218 | 0.2564 |
| Average Road & Asphalt (length in km) | 0 | 3.728 | 11.06 |
| Average Pavement (length in km) | 0 | 1.094 | 2.242 |
| Average pH | 8.618 | 8.562 | 8.416 |
| Average EC (μS/cm) | 159.10 | 493.42 | 1196.30 |
| Average Soil Organic Carbon (SOC) | 0.383 | 0.311 | 0.883 |
| Average Soil Organic Matter (SOM) | 0.659 | 0.534 | 1.521 |
| Average clay | 24 | 21.5 | 23.5 |
| Average silt | 14.5 | 10.5 | 14.5 |
| Average sand | 61.5 | 68 | 62 |
| Texture | Sandy Clay Loam | Sandy Clay Loam | Sandy Clay Loam |
| Average bar ground percentage | 34 | 26 | 20 |
| Average plant cover percentage | 66 | 74 | 79 |
| Average litter cover percentage | 1.36 | 0.5 | 11 |
| Average litter depth | 0.31 | 0.05 | 0.01 |
| Average log percentage | 7.94 | 32.54 | 28.08 |

Within each site, soil samples were collected in triplicate along a diagonal line from the top to 15 cm using a soil auger. All three samples were consolidated into one composite

sample, mixed thoroughly (total weight 1.5–2.0 kg), and sent for analysis to the Soil Laboratory at the College of Food and Agricultural Sciences, King Saud University. A number of physio-chemical characteristics of the soil were measured, including clay, silt, sand, texture, soil reaction (pH), soil electrical conductivity (EC), soil organic carbon (SOC), and soil organic matter (SOM).

We collected vegetation data from ten randomly chosen meter-square quadrats at each site (plant cover, litter cover, litter depth, logs, and bare ground). Plant species were surveyed in April 2019, and specimens were identified at the King Saud University Herbarium within the Department of Botany and Microbiology.

### 2.3. Data Analysis

Across sites and urbanization levels, variation in beetle assemblages and species composition was described based on abundance, species diversity, evenness, Shannon and Simpson diversity indexes, composition, assemblage variability, and indicator species. The mean number of individuals from each species collected from each site was used to measure abundance, while the total number of species recorded was used to measure species richness. All variables were compared using one-way ANOVA tests.

Detrended correspondence analysis (DCA) was used to ordinate the sites against axes based on beetle abundance and species composition [31]. The influence of environmental variables on the beetle assemblages was tested by canonical correspondence analysis (CCA) [32]. By using CANOCO program and the PC-ORD package, we conducted both CCA and DCA analyses. Using the CANOCO program, the CCA was conducted using a forward selection mode, and the significance of each variable was tested sequentially using a Monte Carlo simulation algorithm before adding it to the final model. We included only species present at two or more sites in our DCA and CCA analyses. Models were based on variables that were significant at $p < 0.05$. On CCA triplots, variables are displayed as arrows pointing toward maximum variation, whose length varies according to the rate of change [32]. We examined the similarity among the beetle assemblages using analysis of similarity (ANOSIM, using a nested two-way design). Based on square-root transformed beetle abundance data, ANOSIM was performed on Bray-Curtis similarity matrixes with 999 permutations. We used the program PRIMER version 7.0.17 for the ANOSIM analysis. Indicator species analysis was used to determine the species characteristic of the level of the urbanization gradient [33]. Indicator species analysis was performed using the PC-ORD statistical package.

## 3. Results

### 3.1. Beetle Diversity

The urbanization gradient yielded 2791 individuals representing 94 species from 62 genera and seven families (Figure 2A, Table S1). In the rural habitat, we caught 1170 individuals belonging to 66 species; in the suburban habitat, 712 individuals and 42 species; and in the urban habitat, 802 individuals and 32 species. The family Tenebrionidae dominated the richness (38.3%, 36 spp.) and abundance values (60.1% of the total catch) (Figure 2A). About 54% of the individuals belonged to six species, *Adesmia stoeckleini*, *Mesostena puncticollis* and *Zophosis punctate* (Tenebrionidae), *Anthelephila caeruleipennis* and *Anthicus crinitus* (Anthicidae), and *Bembidion wittmeri* (Carabidae) (Table S1). None of the 94 beetle species occurred at all 15 studied sites. As illustrated by the distribution ranges of some documented species, *M. puncticollis* was recorded at 13 sites and had the highest abundance (14%), whereas *A. cancellata* was found at 11 locations. Twenty-one species, or 22% of the total species recorded, were represented by a single specimen; eight species (8.4%) appeared in all habitats along the urbanization gradient.

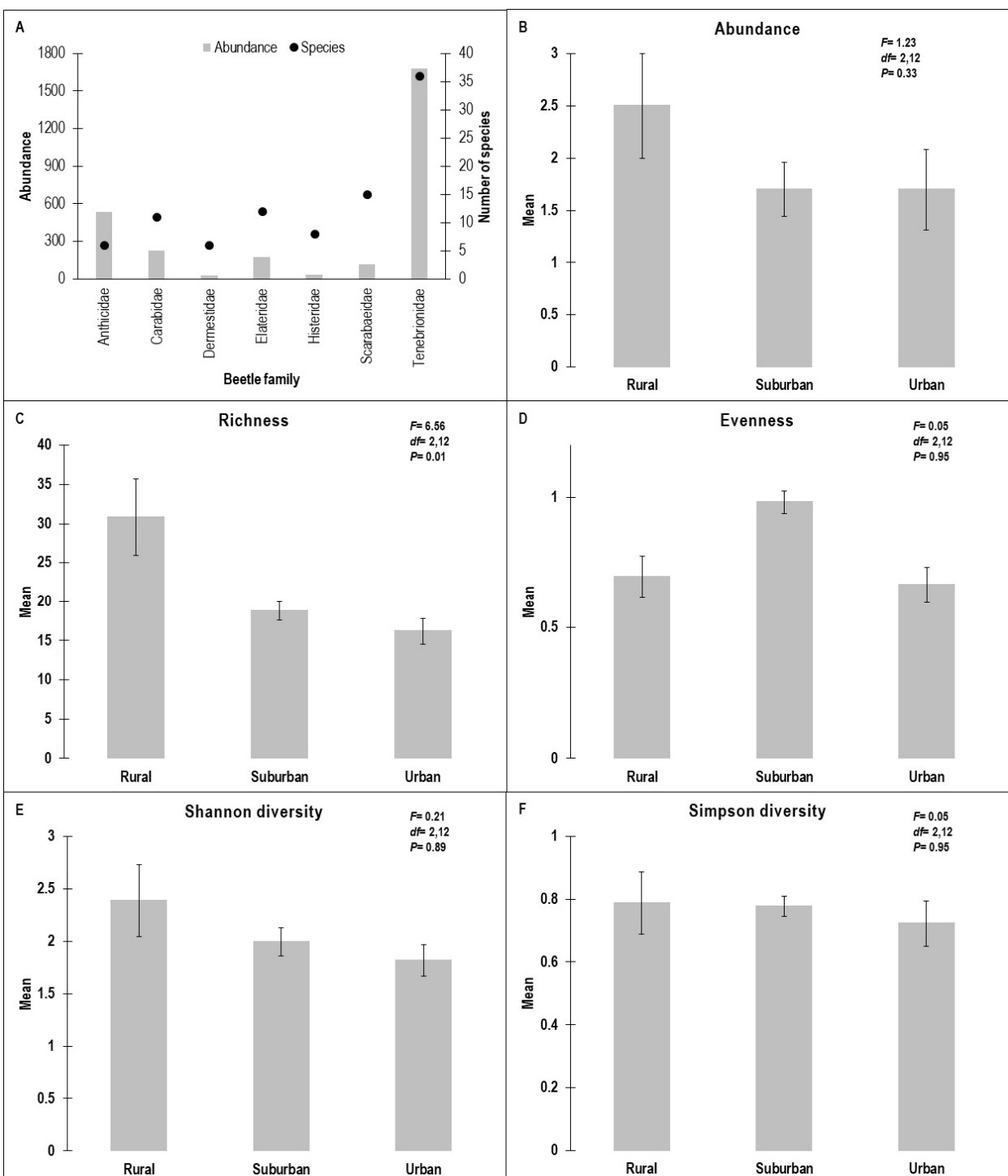

**Figure 2.** The ground-dwelling beetle along the rural-urban gradient in Wadi Hanifa, Saudi Arabia: (**A**) Abundance and richness of collected families; (**B**) mean abundance; (**C**) mean species richness; (**D**) mean evenness; (**E**) mean Shannon diversity; (**F**) mean Simpson diversity.

Rural sites had a significantly higher mean number of species (30.8 species) than suburban (18.8) or urban (16.2) ones (Figure 2C). A gradient of disturbance from rural to urban did not reveal any obvious patterns in abundance, evenness, or Shannon or Simpson diversity (Figure 2B,E,F).

*3.2. Species Composition*

A distinct separation was observed along the urbanization gradient according to DCA analysis. In contrast to the DCA (Figure 3), the CCA produced similar ordination patterns, with most sites remaining in their respective divided groups (Figures 4–7). DCA and CCA yielded small eigenvalue reductions, indicating that this study may have missed other less critical environmental variables. The CCAs considered a variety of environmental variables, including urbanization (3 variables), elevation, soil (7 variables), land cover (5 variables), and flora (43 species). Nevertheless, only urbanization levels (3 variables), soil (1 variable), and land cover (2 variables) were retained in the models, along with four species of flora. The CCAs identified buildings, pavements, roads and asphalt, elevation, soil organic carbon (SOC), soil organic matter (SOM), % litter cover, litter depth, *A. graecorum*, *H. currasavicum*, *L. capitata* and *L. shawii* as essential with beetle species composition/sites (Monte Carlo permutation test, $p < 0.05$) (Table 2, Figures 4–7).

In terms of DCA axis 1 and 2, 15 sites were plotted and clustered into three groups based on urbanization gradients, with eigenvalues of 0.56 and 0.24, respectively (Figure 3). Across axis 1, the urban habitat is separated on the right end and is dominated by the beetles *Endomia lefebvrei*, *Sclerum orientalis*, and *A. caeruleipennis*. At the left end, however, suburban and rural habitats were found. CCA (Figures 4–7) interprets axis 1 as urbanization level, organic contents, and litter, and these variables increase from left to right along axis 1. The rural sites (WHR1–5), along with one suburban site (WHS5), were located on the negative part of axis 2 and were dominated by the beetles *A. stoeckleini*, *Oxycara saudarabica*, and *Scleropatroides* sp. The rest of the suburban sites (WHS1–4) were placed on the positive side of axis 2, with their characteristic beetle species *Pimelia thomasi thomasi*, *Z. punctate*, and *Rhyssemus saudi*. Axis 2 was interpreted as an elevation gradient. Similarity levels among the three urbanization gradient sample groups differed significantly from each other based on ANOSIM ($R = 0.97$, $p = 0.001$).

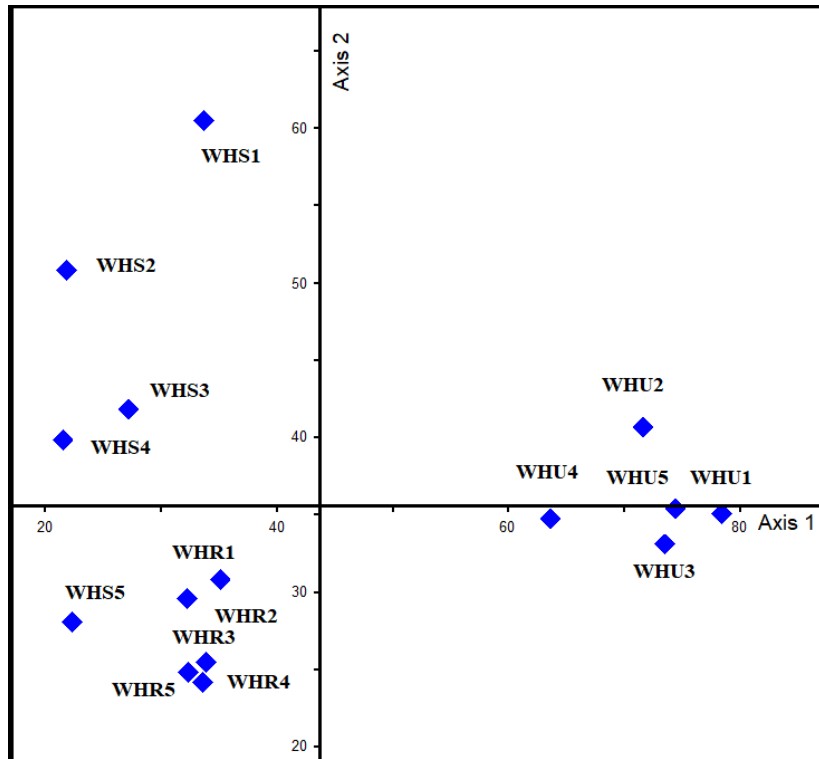

**Figure 3.** Detrended correspondence analysis (DCA) diagram showing the distribution of sites using presence/absence data for the 94 beetle species. WHR, rural habitat; WHS, suburban habitat; WHU, urban habitat.

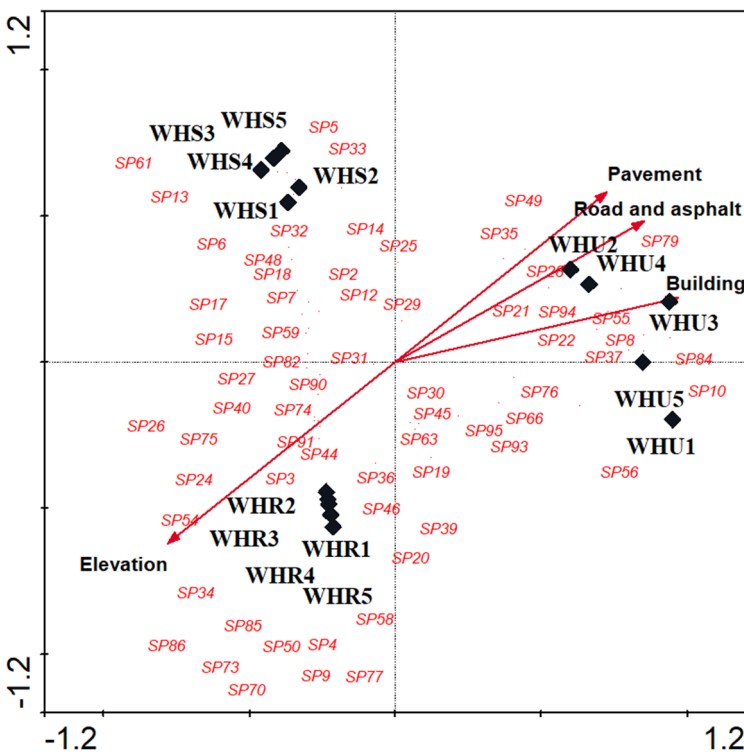

**Figure 4.** Canonical correspondence analyses (CCA) triplot with urbanization level and elevation represented by arrows, different studied sites represented by solid diamonds. And beetle species represented by species code in red. WHR, rural habitat; WHS, suburban habitat; WHU, urban habitat.

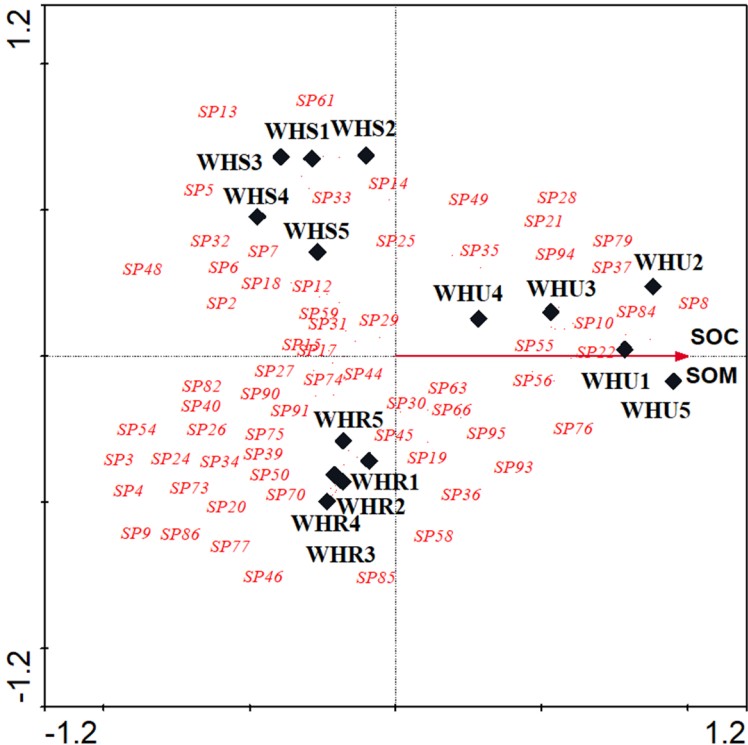

**Figure 5.** Canonical correspondence analyses (CCA) triplot with the essential soil variables (1 of 7) represented by arrows, different studied sites represented by solid diamonds. Beetle species represented by species code in red. SOC, Soil Organic Carbon; SOM, Soil Organic Matter; WHR, rural habitat; WHS, suburban habitat; WHU, urban habitat.

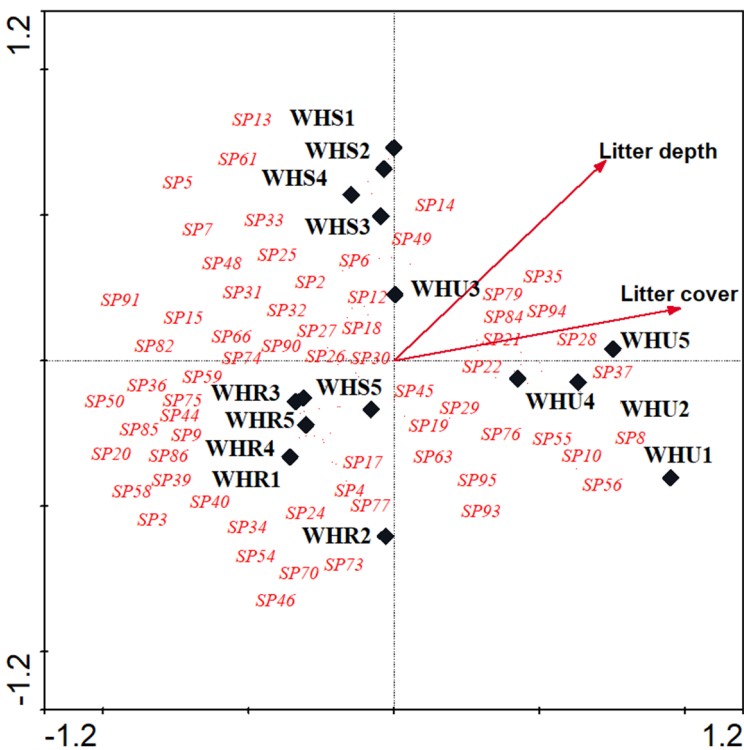

**Figure 6.** Canonical correspondence analyses (CCA) triplot with the essential land cover variables (2 of 5 variables) represented by arrows, different studied sites represented by solid diamonds. beetle species represented by species code in red, WHR, rural habitat; WHS, suburban habitat; WHU, urban habitat.

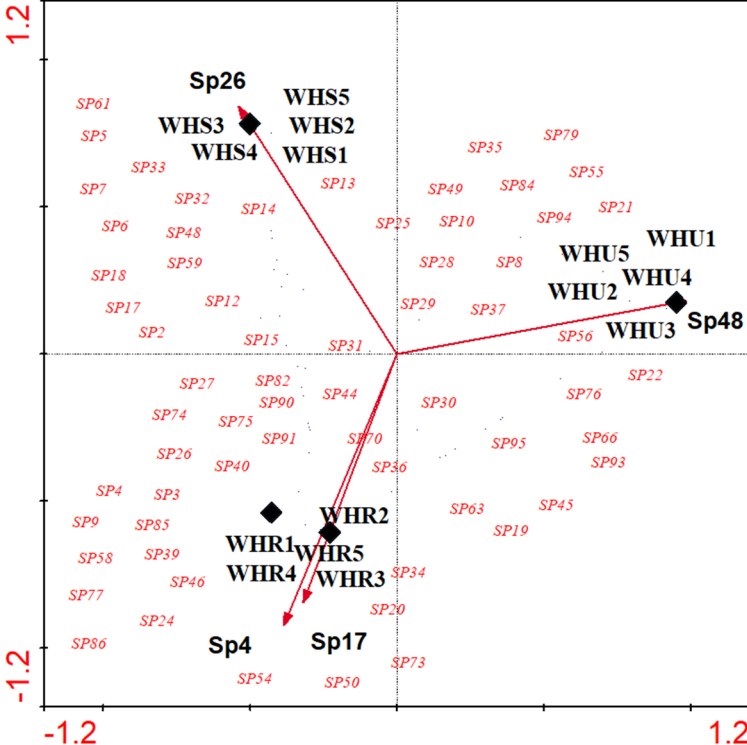

**Figure 7.** Canonical correspondence analyses (CCA) triplot with effective flora species (3 of 43) represented by arrows, different studied sites represented by solid diamonds. beetle species represented by species code in red. Sp4, *Launaea capitata*; Sp17, *Lycium shawii*; Sp26, *Alhagi graecorum*; Sp48, *Heliotropium currasavicum*; WHR, rural habitat; WHS, suburban habitat; WHU, urban habitat.

**Table 2.** Results of the CCA of 15 sites and the Monte Carlo permutation test with *F* and *p* values for the retained investigated environmental variables in the models.

| | Eigenvalue | | | *F* | *p* | Weighted Correlation Matrix | |
|---|---|---|---|---|---|---|---|
| | **Axis 1** | **Axis 2** | **Total** | | | **Axis 1** | **Axis 2** |
| **Urbanization level** | 0.638 | 0.356 | 2.169 | | | | |
| Buildings | | | | 5.18 | 0.0020 | 0.9589 | 0.2135 |
| Pavements | | | | 1.21 | 0.270 | 0.7182 | 0.5657 |
| Roads and asphalt | | | | 0.62 | 0.884 | 0.8437 | 0.4683 |
| **Elevation** | | | | 3.75 | 0.0020 | −0.7707 | −0.6028 |
| **Vegetation cover** | 0.459 | 0.314 | 2.169 | | | | |
| Litter cover | | | | 3.45 | 0.0020 | 0.8676 | 0.1654 |
| Litter depth | | | | 2.74 | 0.0020 | 0.6417 | 0.6300 |
| **Flora** | 0.649 | 0.366 | 2.195 | | | | |
| *Alhagi graecorum* | | | | 3.327 | 0.01 | −0.5362 | 0.8231 |
| *Heliotropium currasavicum* | | | | 5.33 | 0.0020 | 0.9770 | 0.1765 |
| *Launaea capitata* | | | | 3.80 | 0.0020 | −0.3841 | −0.9034 |
| *Lycium shawii* | | | | 1.88 | 0.0260 | −0.3173 | −0.8271 |
| **Soil chemical properties** | 0.568 | 0.385 | 2.195 | | | | |
| Soil Organic Carbon (SOC) | | | | 4.53 | 0.0020 | 0.9420 | 0.0000 |
| Soil Organic Matter (SOM) | | | | 0.67 | 0.80 | 0.9415 | −0.0007 |

### 3.3. Beetle Indicator Species

Based on their Indicator Values, beetle species were divided into three affinity groups (Table 3), namely: (1) species that preferred the urban habitats (e.g., *A. caeruleipennis*, *A. crinitus*, *E. lefebvrei*, *Gonocephalum besnardi*, *G. prolixum*, *G. rusticum*, *Maladera insanabilis*, *Microlestes infuscatus*, *Pentodon algerinus*, *S. orientalis*); (2) species that preferred the suburban habitat (e.g., *Anthrenus malkini*, *G. soricinum*, *P. thomasi thomasi*, *R. saudi*, *Tentyrina deserta deserta*, *Z. punctate*); and (3) species characteristic of the rural habitat (e.g., *A. stoeckleini*, *Akis spinosa*, *Blaps kollari kollari*, *Eremolestes sulcatus*, *Mesolestes quadriguttatus*, *O. saudarabica*, *Scleropatroides* sp.).

**Table 3.** Indicator beetle species identified by indicator species analysis for each urbanization gradient at Wadi Hanifa, Riyadh, Saudi Arabi, with the observed indicator value (IV) and significance level (*p*).

| Family | Species | Gradient | IV | *p* |
|---|---|---|---|---|
| Tenebrionidae | *Adesmia stoeckleini* Koch, 1940 | 0 | 100 | 0.001 |
| Tenebrionidae | *Blaps kollari kollari* Seidlitz, 1896 | 0 | 66.7 | 0.01 |
| Tenebrionidae | *Scleropatroides* sp. | 0 | 66.7 | 0.02 |
| Tenebrionidae | *Oxycara saudarabica* Kaszab, 1979 | 0 | 72.3 | 0.007 |
| Carabidae | *Mesolestes quadriguttatus* (Mateu, 1979) | 0 | 66.7 | 0.02 |
| Carabidae | *Eremolestes sulcatus* (Chaudoir, 1876) | 0 | 66.7 | 0.02 |
| Tenebrionidae | *Akis spinosa* (Linnaeus, 1764) | 0 | 66.7 | 0.016 |
| Tenebrionidae | *Pimelia thomasi thomasi* Blair, 1931 | 1 | 75 | 0.01 |
| Tenebrionidae | *Gonocephalum soricinum* (Reiche & Saulcy, 1857) | 1 | 73.3 | 0.01 |
| Tenebrionidae | *Tentyrina deserta deserta* Kaszab, 1981 | 1 | 54.5 | 0.045 |
| Scarabaeidae | *Rhyssemus saudi* Pittion, 1984 | 1 | 63.8 | 0.035 |
| Tenebrionidae | *Zophosis punctata* Brullé, 1832 | 1 | 72.5 | 0.007 |
| Dermestidae | *Anthrenus malkini* Mroczkowski, 1980 | 1 | 50 | 0.05 |

**Table 3.** *Cont.*

| Family | Species | Gradient | IV | *p* |
|---|---|---|---|---|
| Tenebrionidae | *Gonocephalum besnardi* Kaszab, 1982 | 2 | 60 | 0.02 |
| Tenebrionidae | *Sclerum orientalis* (Fabricius, 1775) | 2 | 80 | 0.006 |
| Anthicidae | *Anthelephila caeruleipennis* (La Ferté-Sénectère, 1847) | 2 | 85.7 | 0.003 |
| Anthicidae | *Anthicus crinitus* LaFerté-Sénectère, 1849 | 2 | 66.7 | 0.011 |
| Scarabaeidae | *Maladera insanabilis* (Brenske, 1894) | 2 | 61 | 0.03 |
| Anthicidae | *Endomia lefebvrei* (LaFerté-Sénectère, 1849) | 2 | 100 | 0.001 |
| Scarabaeidae | *Pentodon algerinus* (Füessly, 1778) | 2 | 60 | 0.03 |
| Carabidae | *Microlestes infuscatus* (Motschulsky, 1859) | 2 | 60 | 0.03 |
| Tenebrionidae | *Gonocephalum prolixum* (Erichson, 1843) | 2 | 60 | 0.02 |
| Tenebrionidae | *Gonocephalum rusticum* (Olivier, 1811) | 2 | 70 | 0.001 |

WHN1–5, WHS5 = 0; WHS1–4 = 1; WHU1–5 = 2.

## 4. Discussion

Among the most important approaches to studying the effects of urbanization is the rural-urban gradient methodology. Nevertheless, most studies tend to be more varied and oversimplified when presuming linear patterns of change or difference [34]. In contrast, the rural-urban gradients tend to exhibit a heterogeneous and complex relationships. Urbanization leads to habitat degradation, fragmentation, and loss, often associated with biodiversity loss and significant changes in species composition [16,18]. A significant impact of urbanization is the promotion of urban adapters and exploiters, which are usually exotic generalist species [16,19]. Invertebrates are significantly impacted by urban gradients due to their capacity to capture a wide range of urban impacts [20,21]. Several factors contribute to the importance of the study area, including its status as a prominent natural landmark, its status as the largest and most important valley in Riyadh, and its position in the middle of the Najd Plateau of Saudi Arabia [29,35,36].

### 4.1. Beetle Diversity

The preponderance of the family Tenebrionidae at these sites (38.3% of total species, 60.1% of total catch) characterizes the insect fauna in arid and semi-arid environments. This result and those found previously in arid and semi-arid environments [26,37–41] highlight a strong dominance of darkling beetles (Tenebrionidae), which indicates a highly adaptable beetle fauna. All rural, suburban sites and 1–3 urban sites were dominated by tenebrionid species such as *A. cancellate*, *M. puncticollis*, and *Trachyderma philistina*. In rural and suburban habitats (98.9%), they were more abundant than in urban habitats (1.1%). A decline in their number in urban habitats may be attributed to the alteration of microhabitats as a result of urbanization [21,42,43]. Another factor that may affect habitat choice [44] is the higher temperature resulting from urbanization [45,46]. As urbanization progresses, the number of species-rich beetle habitats changes significantly, with rural habitats having a higher species richness than suburban and urban habitats. As surprising as it might seem, species abundance, evenness, and diversity were invariant along urbanization gradients despite the change in species richness. A possible explanation for this apparent contradiction is the diverse reactions of beetle species to urbanization and their sensitivity to environmental changes. Some evidence suggests that poor dispersing species may have limited environmental tolerances and may be particularly sensitive to urbanization (e.g., *Adoretus granulifrons*, *Micipsa arabica*, *Mycetocharina vanharteni*, *Schizonycha* sp., and *Sepidium tricuspidatum*). Urban habitats provide a lower availability of resources (dung) than rural habitats, which limits the diversity of coprophilous species (species (e.g., some members of Glaresidae, Histeridae, and Scarabaeidae) [47]. The urbanization-induced conditions may

also benefit generalists and widespread species (e.g., *G. rusticum* and *T. philistina*) and also generalist species (e.g., *M. puncticollis* and *Thriptera kraatzi*) [2,48].

### 4.2. Species Composition

Overall diversity along the urbanization gradient generally displayed heterogeneous patterns [49]. Species, abundance, richness, and diversity of beetles showed one of following three patterns. The first decreased along the rural-urban gradient [22,50–54]. Second, some species exhibit higher species richness and diversity in urban habitats than in rural areas [55–59]. Finally, there are those species that show no significant differences between urban and rural habitats [7,42,60–64]. Based on these patterns and our results, it seems that species richness, abundance, or diversity are not entirely appropriate parameters for assessing the effects of urbanization on soil and ground-dwelling beetles [2]. Species identity and associated natural history offer greater insight into a beetle composition in these habitats than species richness or diversity alone [41,65,66].

A range of factors, such as local urbanization intensity, edaphic factors [67–69], vegetation cover, and flora [42,70–74], contribute to variations in the distribution and composition of beetle species associations along Wadi Hanifa. As revealed by CCA, the most significant factors determining the distribution of beetles are urbanization intensities, elevation, soil organic contents, land cover, and flora. The DCA has determined that the beetle list falls into three distinct clusters: urban, suburban, and rural. Each cluster contains a different assemblage of beetles (<9% of species shared). ANOSIM analysis confirmed this grouping (R = 0.97, *p* = 0.001), which aligns with previous reports on ant assemblages in the same area [28]. This was associated with the increasing density of buildings, soil organic carbon, percentage and depth of litter cover, as well as the presence of *H. currasavicum*, all of which were strongly correlated with axis 1. Also associated with small-sized (e.g., *Gonocephalum* spp., *M. infuscatus,* and *S. orientalis*) and scavenger and opportunistic predator beetle species (e.g., *A. caeruleipennis*, *A. crinitus*, and *E. lefebvrei*). Unlike axis 1, the second axis was associated with elevation and the predominant flora of *A. graecorum*, *L. capitata*, and *L. shawii*. In rural habitats, sites with a high elevation and dominant flora of *L. capitata* and *L. shawii* were associated with large, xerophilic species (e.g., *Anthia duodecimguttata*, *A. spinosa*, and *B. kollari kollari*), predators (e.g., *E. sulcatus M. quadriguttatus*), and detritivores (e.g., *Scleropatroides* sp.). There was a correlation between beetle assemblage structure and rural plants (*L. capitata*, *L. shawii*, and *M. parviflora*), suburban plants (*Casuarina equisetifolia* and *A. graecorum*), and urban plants (*Panicum coloratum*, *Atriplex nummularia*, and *H. currasavicum*). It appears that soil moisture and nutrients were similar for these groups [75,76]. The composition of beetle communities was generally influenced by the tree canopy and dense herb cover, which in turn impacts soil moisture, solar radiation, microclimate conditions, as well as the type of prey available [42,77,78].

Lack of management contributes to an increase in litter and logs [42]. In unmanaged urban sites within Wadi Hanifa, an increase in litter cover and depth has been observed due to the conversion of native flora to perennial vegetation and introduction of irrigation. As a result, a significant influence of leaf litter cover and depth was observed in our study on the composition of beetle communities. Litter layers are argued to sustain a significant portion of beetle diversity by creating favorable microenvironments [79–81] for prey [72] or larger polyphagous and generalist beetle species [58] and enhancing egg and larval development [56]. Nevertheless, this study found that species richness decreased in conjunction with litter percentage and depth increases, which is consistent with the finding of Guillemain et al. [82], Molnar et al. [83], and Fuller et al. [84].

The organic content of soil (SOC, SOM) also plays a vital role as critical factors in the soil [63,85,86]. Subsequently, the insects inhabiting the soil significantly influenced the number and composition of trapped beetle species [67,87–89]. The CCA analysis indicated that both SOC and SOM were higher at urban sites than at suburban and rural sites, which agrees with Asabere et al. [15], who concluded that dumping inorganic and organic materials and decaying plant material resulted in increased SOM and SOC in

urban areas. This was reflected by the occurrence of a large number of small-sized beetles (e.g., *Anthelephila caeruleipennis*, *A. crinitus*, and *Endomia lefebvrei*). These indicator beetles correlated positively with SCO and SOM in urban habitats, whereas suburban and rural indicator beetles showed negative correlations with SCO and SOM.

The construction of houses in urbanized communities leads to homogenized landscapes and uniform disturbance on all sites where topsoil is removed and replaced with pavement. The rigid surfaces (such as buildings, roads, and paving) not only seal off large soil areas but also enhance the proliferation of ruderal plant species. As a result, many soils and ground-dwelling animal habitats are altered [21,42,43,90,91]. As one moves from the heart of Riyadh City to the south, the buildings, pavements, roads, and asphalt represent a gradient of urbanization intensity along Wadi Hanifa. Our urban study sites gradually became overrun with exotic plants such as *A. nummularia*, *Pennisetum setaceum*, and *Phragmites australis* as native vegetation was gradually replaced. As a result of human modification of habitats, certain types of beetles are more likely to survive, such as opportunistic predators (e.g., members of the family Anthicidae).

*4.3. Beetle Indicator Species*

According to New [24], the variation within each habitat can be explained by shifts in highly abundant species. Across the rural-urban gradient, beetle indicators differed in terms of their ecological factors. According to the present findings, urban sites that experienced the most disturbance differed from rural and suburban sites in composition. A total of 23 characteristic beetles have been identified, and these play an essential role in determining the structure of the beetle assemblage, as shown by the ordination results. Urban habitats have a high content of indicator species (10 species), perhaps due to the high number of plants and trees [73,74]. The perennial vegetation and the irrigation in unmanaged urban areas may act as source habitat for herbivores (e.g., *M. insanabilis*, *P. algerinus*, and *Gonocephalum* spp.) and opportunistic predators (e.g., members of Anthicidae) [58]. With increasing levels of anthropogenic disturbance, urban habitats can retain some species from the natural habitats (e.g., *M. puncticollis*, *T. kraatzi*, and *T. philistina*). However, their richness and abundance will also decrease [62]. The occurrence of native, habitat-specific beetles (e.g., *B. wittmeri* and members of family Elateridae) in rural areas tend to be higher than in urban areas [42,92,93].

**5. Conclusions**

In response to urbanization, beetle species composition has significantly changed, resulting in heterogeneous diversity patterns [49] and subsequent loss of native species [16,18]. The study found that local edaphic factors, vegetation cover, and flora can drive changes in the number of species and composition of beetles along rural-urban gradients. In arid ecosystems, darkling beetles (Tenebrionidae) are conspicuous components that may help to illustrate biodiversity changes in response to various environmental changes. Several interactions and responses have been observed due to different levels of urbanization and environmental factors, including changes in the small species, native species, generalists, coprophilous species, and opportunistic species. Our study concludes that species richness, abundance, and diversity are not entirely appropriate parameters to assess the impact of urbanization on soil and ground-dwelling insects [2,28]. In fact, the composition of beetles is quite fundamental, and studying species identity and associated natural history will provide a much deeper understanding of environmental changes than focusing solely on species richness and diversity [41,65,66]. It is also necessary to investigate the response of insect-feeding guilds to the microhabitat alterations due to urbanization in Saudi Arabia, as these issues need to be addressed in future research.

**Supplementary Materials:** The following supporting information can be downloaded at: https://www.mdpi.com/article/10.3390/d15020303/s1, Table S1: List of ground-dwelling beetles and their abundance along the rural-urban gradient in Wadi Hanifa, Riyadh, Saudi Arabia.

**Author Contributions:** Conceptualization, M.S.A.-D., M.R.S., J.D.M. and G.M.O.; methodology, M.S.A.-D., J.D.M., M.K.A.-S., A.S.A., H.M.A. and G.M.O.; validation, M.R.S., A.M.S., A.S.A. and H.M.A.; formal analysis, M.S.A.-D., J.D.M. and G.M.O.; investigation, M.S.A.-D., M.R.S., M.K.A.-S., A.M.S., A.S.A. and H.M.A.; resources, A.M.S.; data curation, M.SA., M.R.S. and G.M.O.; writing—original draft preparation, M.S.A.-D. and G.M.O.; writing—review and editing, all authors; visualization, M.S.A.-D., J.D.M. and G.M.O.; supervision, M.K.A.-S. and M.R.S.; project administration, H.M.A.; funding acquisition, M.S.A.-D. All authors have read and agreed to the published version of the manuscript.

**Funding:** This research was funded by NSTIP strategic technologies programs, project number [12-ENV2804–02], The King Abdulaziz City for Science and Technology, National Plan for Science, Technology and Innovation.

**Institutional Review Board Statement:** Not applicable.

**Data Availability Statement:** The data presented in this study are available in Supplementary Material here.

**Acknowledgments:** The authors thank the King Saud University Museum of Arthropods team for assisting in the field work and sorting the collected specimens. We like to extend our thanks to soil laboratory staff of the Soil Department, College of Food and Agricultural Sciences, King Saud University, for their help in soil analyses. We are grateful to Jacob Thomas, Herbarium, Department of Botany & Microbiology, College of Science, King Saud University, Saudi Arabia, for plant identification.

**Conflicts of Interest:** The authors declare no conflict of interest.

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
