# Peer review of "Does a Rural-Urban Gradient Affect Beetle Assemblages in an Arid Ecosystem?"

_diversity, doi:10.3390/d15020303_

Round 1
Reviewer 1 Report
The research shows the effect of a gradient of urbanization in beetle communities in an arid region. Generally, it is properly introduced. The methods are correct, the number of sample sufficient and results are well presented. Some edition in the English would improve the quality of the manuscript (particularly in the discussion).
Few comments follows:
Tittle
I suggest replacing arthropod by beetle.
Abstract
Line 26. “In Riyadh…” I suggest indicating if Riyadh is a municipaly/city/region for non-Arabian readers, i.e., In Riyadh city.
Line 29. Idem for Wadi Hanifa (all across the manuscript with toponyms).
Line 31. I suggest writing the extended form of DCA, CCA
Line 35. …plant species such as…
Line 36. Ten species in our study were associated with
Introduction
Line 49. “…both at scale and within individual sites” The meaning is not clear. Please, clarify.
Line 51-52. “...Additionally, disturbed natural habitats often become non-permanent environments such as abandoned sites characterized by altered chemical and physical properties of soil…” what do you mean by non-permanent environments and abandoned sites? Please, be more specific.
Line 64-65. “…their diverse life histories provide insight into the composition of metropolitan fauna” How the diversity of life histories of arthropods provides insights about all the metropolitan fauna? Please, clarify.
Line 75. “Despite this fact…” Should not be more appropriated “for that reason”?
Line 77. Please, state the goals of your work at the end of the introduction.
Material and Methods
Line 81. Is “Riyadh” a region?
Line 84-84. Capitalize Wide and indicate that it is a river.
Figure 1. I recommend showing the Wadi River, the Riyadh city and the desert in the map. How important is the river for this research? Were the sampled sites a similar distance from the river?
Line 91. “…but it does not rain from June to September…” I suggest: “ The dry season, with the absence of rain, occurs from June to September”
Line 92. “Haysiah Dam” is this a city/village?
Line 99-105. Which area is this paragraph referring to? Suburban?
Line 153. Remove “This is a table. Tables should be placed in the main text near to the first time they are cited.” and include the table 1 caption.
Line 126. Please, indicate all characteristic of the pitfall trap in order to enable the replicability of the experience (material, diameter and depth of the recipient).
Line 131. Please indicate the concentration of the solution.
Line 132. Please, specify the months/date of sampling.
Line 132-134. I suggest “The identification of the collected beetle species was conducted at the King Saud University Museum of Arthropods (KSMA), Department of Plant Protection, College of Food and Agricultural Sciences”
Line 135-136. “To measure the level of urbanization, we used the total amount of built-up area as a 135 proxy. The total amount of built-up area was used to reflect the urbanization level.” Both sentences indicate the same information.
Results
Were the different among sample months evaluated?
Line 209. “Monte” Correct typo.
Line 232. I suggest to perform a post-hoc test for richness to indicate the difference among the urbanization levels. Correct typo in “abundance” in the panel B.
Table 2 and 3. Uniformize the number of decimal digits.
Table 3. Some decimals are indicated with comma instead of point.
Discussion
Line 268. “most studies tend to be more varied and oversimplified when presuming linear patterns of change or difference” This needs some clarification.
Line 272. “adapters and exploiters” What do you mean?
Line 274-277. “Several factors contribute to the importance of the study area, including its status as a prominent natural landmark, its status as the largest and most important valley in Riyadh, and its position in the middle of the Najd Plateau of Saudi Arabia [29,35,36].” I suggest to move this to the introduction.
Line 281. Remove “both”
Line 282-283. “which indicates a highly adaptable beetle fauna. The high species richness (38.3% of total species) and abundance (60.1% of total catch) are due to this adaptation.” In the way this is written, it is redundant. Please, rephrase.
Line 284-285. “All rural and suburban sites and 1-3 urban sites were dominated by tenebrionid species such as A. cancellate, M. puncticollis, and Trachyderma philistina were found to be the most” This sentence is grammatically incorrect. Please, rephrase.
Line 306. I suggest “showed one of following three patterns.
Line 307. “The first are those decrease along the rural-urban gradient” This sentence contains several grammar mistakes. Please, revise.
Line 306-314. Are you speaking about your research or about other research? It is not clear. Please clarify.
Line 332. There was?
Line 335. Do you mean that the “soil moisture and nutrients” effects were similar for those groups? Please, clarify.
Line 336. was generally influenced.
Line 352. “significantly influenced”? The CCA…?
Line 356-259. Can be these sentences joined?
Line 360-370. This paragraph is difficult to follow. Please, rephrase.
Author Response
We appreciate the generally positive comments from the reviewer, which have substantially improved our paper. Our responses to the comments are detailed in the attached file.

Reviewer 2 Report
This study comprised field surveys in different urbanization gradients as well as ecological indices to demonstrate the influence of urbanization on beetle assemblages. Overall, the work is well-written and simple to understand. The findings are particularly important since they show a shift in the composition of beetle species at different degrees of urbanization and indicate a potential loss of biodiversity as the urbanization process accelerates.
I'd like to congratulate the authors; the work is easy to read, includes appropriate methods and statistical analysis, and conveys the primary idea clearly. I'd only want to suggest a higher image resolution.
Best wishes.
Author Response
We are grateful for the reviewer's generally positive comments.